# Work-related musculoskeletal disorder and its associated factors among bank workers in Ethiopia: A systematic review and meta-analysis

Abebe Kassa Geto [1]*, Hussien Mekonnen[2], Tesfalem Tilahun Yemane[2], Endalew Minwuye Andargie[3], Birhanu Sewunet[4], Tarikuwa Natnael[4], Chala Daba[4]

1 Department of Public Health, College of Health Sciences, Woldia University, Woldia, Ethiopia, 2 Department of Nursing and Midwifery, Dessie Health Science College, Dessie, Ethiopia, 3 Department of Health Service Management, School of Public Health, Asrat Woldeyes Health Science Campus, Debre Berhan University, Debre Berhan, Ethiopia, 4 Department of Environmental Health, College of Medicine and Health Sciences, Wollo University, Dessie, Ethiopia

* abebekassa2129@gmail.com

## Abstract

### Introduction

Work-related musculoskeletal disorders are among the major global public health problems and contributors to disability and workers' absence in occupational areas which certainly disrupts work productivity and expected results. In Ethiopia, different studies investigated work-related musculoskeletal disorders. However, the findings were not consistent and conclusive enough, and there is no nationwide data representing this growing public health concern. This in turn hinders the efforts of intervention activities. Therefore, this study aimed to estimate the pooled prevalence of and factors associated with work-related musculoskeletal disorder among bank workers in Ethiopia.

### Methods

To retrieve all the relevant studies, international databases such as PubMed/MED-LINE, CINAHL, LIVIVO, African Journals Online, African Index Medicus (AIM), HINARI, Science Direct, Cochrane Library, Google Scholar, Semantic Scholar, and Google were used. The Preferred Reporting Items for Systematic Reviews and Meta-Analysis (PRISMA) guideline was followed for this study. The extracted data were analyzed using STATA 17 software. The heterogeneity of the included studies was determined by the Higgs $I^2$ statistics. With a 95% confidence interval, this meta-analysis with the random-effects model was carried out to determine the pooled prevalence of work-related musculoskeletal disorder.

**Data availability statement:** All relevant data are within the paper and its Supporting Information files.

**Funding:** The author(s) received no specific funding for this work.

**Competing interests:** The authors have declared that no competing interests exist.

**Abbreviations:** BMI, Body Mass Index; DALYs, Disability-Adjusted Life Years; YLD, Years Lived with Disability; MSDs, Musculoskeletal Disorders; POR, Pooled Odds Ratio; PRISMA, Preferred Reporting Items for Systematic Reviews and Meta-Analysis; WMSDs, Work-related Musculoskeletal Disorders.

## Result

In this meta-analysis, eight articles in total with 3399 study participants were included. The overall pooled prevalence of work-related musculoskeletal disorder was 57.41% (95%; CI: 38.87%, 75.95%; $I^2 = 99.4\%$; P = 0.000). Gender, job stress, physical activity, and work experience were found to be factors significantly associated with work-related musculoskeletal disorder.

## Conclusion

A high prevalence of work-related musculoskeletal disorder among bank workers in Ethiopia was recorded. This underscores the importance of implementing effective intervention measures such as job rotation, job enrichment, extended breaks, mitigating negative social factors, and establishing physical exercise facilities to address the underlying issues.

## Introduction

Work-related musculoskeletal disorders (WMSDs) are musculoskeletal disorders resulting from workplace exposures. They largely affect body parts such as muscles, bones, fascia, joints and tendons, ligaments, nerves, or circulation systems that priorly are induced or worsened by work and the circumstances of the occupational areas [1–4]. Musculoskeletal disorders (MSDs) are impairments of body structures that are caused or worsened by poor fitness, and poor habits of health, but a larger proportion of MSDs are caused by exposures to physical work [3] and this problem occurs secondary to cumulative trauma and suddenly happened injuries, with the former being the most frequent mechanism behind WMSDs [5,6].

WMSDs in the working environment are a major public health concern across the globe. They are common occupational area health problems mainly manifested by a range of symptoms like pain, aches, and discomfort in different parts of the body region. Despite WMSD has been stated as one of the major contributors to disability and workers' absence in the workplace which certainly disrupts work productivity and expected work results, it is responsible for multiple work interruptions/stoppages and substantial direct and indirect costs [1,2,5,7]. Employees of many occupations live with a health burden linked with the disabling musculoskeletal pain and injuries of an occupational-related cause, collectively known as WMSDs [8]. Banks are one of the occupational areas where employees are subjected to various physical demands, prolonged sitting or standing and awkward postures, long working hours, a repetitive tasks in front of computers without having adequate rest and recovery time which may lead to WMSDs [9,10].

A systematic review and meta-analysis on the global prevalence of work-related musculoskeletal disorders among physiotherapists found that WRMSD pooled prevalence in neck, upper back, lower back, shoulders, elbows, wrists/hands, thumb, hips/thighs, knees/legs, and ankles/feet was 26.4%, 17.7%, 40.1%, 20.8%, 7.0%, 18.1%,

35.4%, 7.0%, 13.0%, and 5% respectively [5]. Another systematic review and meta-analysis conducted among African school teachers revealed that the overall estimated pooled prevalence of low back pain was found to be 59% [11]. Another study in Ethiopia found that the pooled prevalence of occupational-related pain on elbow, wrist/hand, knee/leg, foot/ankle, and hip/thig in the previous one year was 19.7%, 24.2%, 25.0%, 20.2%, and 15.5%, respectively [12].

A study in Pakistan reported that back pain was one of the main symptoms causing burnout in bank employees [10]. Working with a computer poses awkward postures that are continually and forcefully maintained and this subsequent change from normal sitting postures while using a computer has been noticed and influences the development of musculoskeletal system pain, back and neck pains being more common [13].

According to evidence from the recent global burden of disease (GBD), 31.3% of the global disability-adjusted life years (DALYs) in 2021 were due to years lived with disability (YLDs). Of which, low back pain (LBP) was the leading Level 3 cause of YLDs with 70.2 million YLDs, and other MSDs combined were the fifth-ranked causes of YLDs with 43 million YLDs [14]. Many of the work-related problems are preventable. However, there are about 2.9 million work-related deaths globally every year. Moreover, 6% of all the deaths in the world were attributed to be work-related. In every single day, over 7,500 people die following workplace accidents [15]. Globally, there were 322.75 million incident cases, 117,540 deaths, and 150.08 million DALYs of MSDs in 2019 [16].

The prevalence of WMSD in Africa is estimated to range from 15% to 93.5% [17] and it remains less focused and overlooked in low-middle-income countries (LMICs), particularly in Ethiopia due to the focus on more pressing and life-threatening health issues like infectious diseases and non-communicable diseases (NCDs) [18]. WMSD is a multi-factorial problem having many possible causes and determining predictors for it is a very difficult task. Age, female gender, physical activity, socioeconomic status, and increased body mass index (BMI) were identified as predictors of WMSD according to different studies [19–21]. Moreover, predictors such as job dissatisfaction, limited social support from workplace partners and supervisors, job stress, sleeplessness, and depression were also depicted as predictors associated with WMSD in the literature [22].

Although studies have been carried out on WMSDs among bank workers across various areas of the country Ethiopia [23–30], the findings have not been consistent with the prevalence ranging from 11.7% [30] to 77.6% [23] and conclusive, which could hamper the assessments of ongoing intervention efforts and activities. Additionally, no study provides countrywide evidence of the pooled prevalence of work-related musculoskeletal disorder among bank workers in Ethiopia. Therefore, this systematic review and meta-analysis aimed to estimate the pooled prevalence of work-related musculoskeletal disorder and identify associated factors among bank workers in Ethiopia. The results of this systematic review and meta-analysis will help policy-makers, healthcare planners, and other concerned bodies to plan and implement strategies to prevent, control, and reduce the impacts of WMSD.

## Materials and methods

### Study registration

A clear protocol for this systematic review and meta-analysis was registered on the International Prospective Register of Systematic Reviews (PROSPERO) database. The registration number is CRD42023441157.

### Search strategy and study selection

In this systematic review and meta-analysis, published and unpublished studies were searched from different electronic databases such as PubMed/MEDLINE, Science Direct, Cochrane Library, LIVIVO, CINAHL, African Journals Online, Web of Science, African Index Medicus (AIM), HINARI, Semantic Scholar, Google, and Google Scholar. Besides, gray literature were also identified from digital libraries and repositories of different universities. The search was carried out using the following keywords: "musculoskeletal disorder", musculoskeletal disease", "musculoskeletal pain", "orthopedic

disorder", "musculoskeletal problem", "musculoskeletal injur*", "musculoskeletal condition*", banker*, "bank employe*", "bank worker*", "bank staff*", "bank officer*", "bank professional*", "bank accountant*", "bank manager*", "bank cashier*", "bank financer*", "bank teller*", "associated factor*", "risk factor*", determinant*, predictor*, precursor*, cause*, "factors associated" and "Ethiopia." Using the Boolean operators "AND" or "OR" as appropriate, all keywords were combined. The search was carried-out until 19 January 2024 by four independent authors (AKG, CD, HM, and BS). The articles searched from the selected electronic databases were transferred to the Endnote version 8 software and all duplicate files were excluded. Selecting all the relevant articles followed the Preferred Reporting Items for Systematic Reviews and Meta-Analysis (PRISMA) guidelines [31] (S1 Table).

## Inclusion criteria

**Population**: All the studies conducted on work-related musculoskeletal disorder among bank workers in Ethiopia.
**Exposure**: Bank workers that exhibited WMSD
**Comparison**: Workers that didn't exhibit WMSD
**Outcome**: Studies assessed WMSD at least at one of the body regions (neck, shoulder, elbow, wrist/hand, upper back, lower back, hip/thigh, knee, and foot/ankle) as the primary outcome in the last 12 months.
**Study setting**: Institution-based studies.
**Study design**: All gray literatures and published studies following cross-sectional, cohort, and case-control study designs were included in this systematic review and meta-analysis.
**Publication**: Published and unpublished studies.
**Country**: All the relevant studies conducted in Ethiopia.
**Language**: Only studies reported in the English language were included.

## Exclusion criteria

Studies with no full text, unidentified reports, abstracts, editorials, irretrievable studies, letters, qualitative studies, and studies that did not report the outcome of interest (WMSD) were excluded.

## Outcome assessment

The main outcome of this study was to determine the pooled prevalence of work-related musculoskeletal disorder among bank workers. This is determined by multiplying by 100 after the number of study participants (the numerator) who had WMSD at least at one of their body regions (neck, shoulder, elbow, wrist/hand, upper back, lower back, hip/thigh, knee, and foot/ankle) was divided by the actual sample size (the denominator). In addition to this, the systematic review and meta-analysis aimed at identifying the factors associated with WMSD in log odds ratio form.

## Operational definition

Work-related musculoskeletal disorders: These are the developed impairments on the muscles, tendons, fascia, ligaments, joints, nerves, bones, or circulation systems of the body that are caused, induced, or worsen by work and the circumstances of its performance in workplaces or occupational settings at least at one part of the body organs (neck, shoulder, elbow, wrist/hand, upper back, lower back, hip/thigh, knee and foot/ankle) [3,4,32].

## Data extraction procedure, study quality, and risk of bias assessment

A standard data extraction template consisting of several details of the study such as author name, region, year of publication, study design, response rate, quality score, bank type, methods of data collection, and prevalence was prepared. Three independent authors (AKG, CD, and BS) undertook all the required data extraction activities. Duplicate articles

were removed after the relevant articles for inclusion were carefully screened by three different reviewers (AKG, CD, and HM). Using the Joana Brigg Institute (JBI) checklist of critical appraisal for cross-sectional studies, the quality of each article was critically evaluated [33] (S2 Table). With scores measured on a scale of 100%, the quality of each article was independently assessed by two different authors (AKG and CD) and a third author (HM) was ready to address and resolve any discrepancies encountered during the quality assessment.

For further analysis, articles having a quality score of 50% and above were included [34,35].

The mean score was computed from the evaluations of all the reviewers to address and resolve any differences in the case of any discrepancies encountered during the quality assessment. Missing data were successfully handled by providing critical care prior to data extraction and giving a special emphasis during data extraction along with the study quality assessment result.

### Statistical analysis procedures

Data was analyzed using STATA (Corporation, College Station, Texas, USA) version 17 software after the extracted data were imported. Heterogeneity within the included studies was assessed using the Higgs $I^2$ test, with values of 75%, 50%, and 25% showing high, moderate, and low levels of heterogeneity respectively [36]. In this meta-analysis, high heterogeneity was observed across the studies included ($I^2 = 99.4\%$; $P = 0.000$). The issue of heterogeneity was addressed by undertaking sensitivity test and subgroup analysis. With a 95% confidence interval, Der Simonian and Liard's [37] method of random-effects model was used to determine the pooled prevalence of work-related musculoskeletal disorder among bank workers since a random effects model can provide more accurate estimates when there is substantial heterogeneity between studies. The odds ratio was computed to show the strength of the association between WMSD (the outcome variable) among bank workers and its risk factors.

To present the pooled prevalence of work-related musculoskeletal disorder, a forest plot was used. To determine the influence of an individual study on the pooled prevalence estimate of WMSD, a sensitivity analysis was performed. Subgroup analysis was also computed to identify the possible sources of heterogeneity based on the year of publication (before 2022 and 2022 and after), study regions (Amhara region and other regions), method of data collection (self-administration and interview), and sample size category (425 and above and lower than 425). Besides, the presence of potential publication bias was determined by using a funnel plot and Egger's test [38].

### Results

### Study selection

Overall, 2896 studies were identified from an electronic database and reference searching. Endnote 8 was used as a reference manager. Thirty-three duplicated articles were removed. The total number of articles excluded based on their titles and abstracts due to the failure to meet the criteria of inclusion was 2844. Besides, 11 articles were excluded as they failed to meet quality assessment methods and did not report the outcomes of interest. In this meta-analysis, a total of 8 full-text articles were included to estimate the pooled prevalence of WMSD by following the PRISMA guideline (Fig 1).

### Characteristics of the included studies

In this meta-analysis, eight cross-sectional studies with a total of 3399 study participants were included. The highest prevalence of WMSD was found to be 77.6% [23] and the lowest prevalence was 11.7% [30] among the included studies. Regarding the study region, four of the included studies were conducted in the Amhara region [26,27,29,30], and the remaining four studies were conducted in Oromia [25], Tigray [24], Addis Ababa [23], and SNNPR [28]. Four [24,25,27,30] of the included articles used interviews as a data collection method, and the other four [23,26,28,29] used self-administration (Table 1).

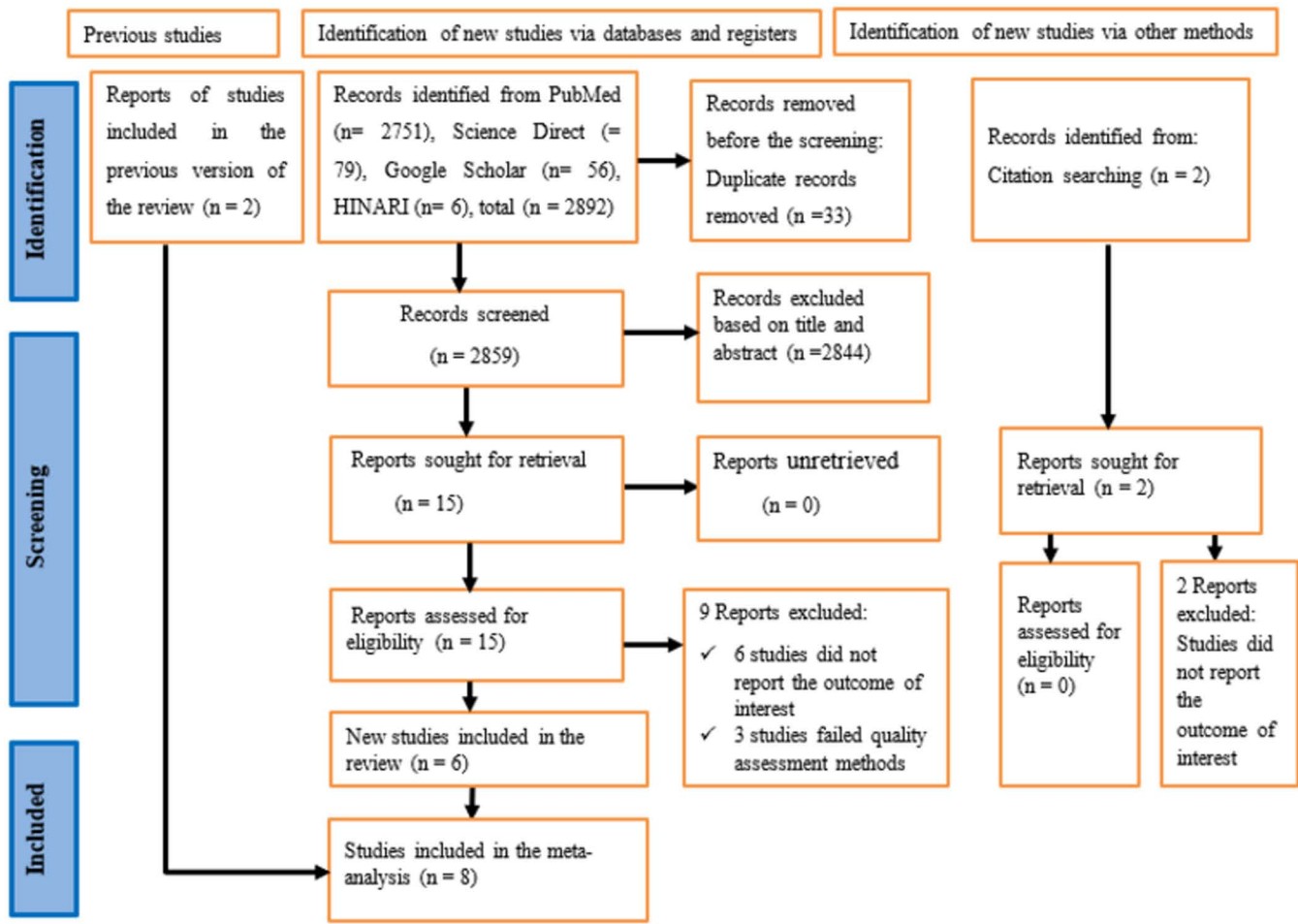

**Fig 1. A PRISMA flow chart showing study selection for systematic review and meta-analysis on the prevalence of WMSD and its associated factors among bank workers in Ethiopia, 2024.**

## Meta-analysis

**Pooled prevalence of work-related musculoskeletal disorder (WMSD).** To determine the pooled prevalence of work-related musculoskeletal disorder in this meta-analysis, eight articles were included. The pooled prevalence of work-related musculoskeletal disorder among bank workers in Ethiopia was found to be 57.41% (95%; CI: 38.87%, 75.95%).

A random effects model was employed to estimate the pooled prevalence of WMSD following the high level of heterogeneity among the included studies ($I^2 = 99.4\%$; $P < 0.000$) (Fig 2).

**Test for publication bias.** The symmetrical distribution of the included studies shown by the funnel plot indicated that there was no publication bias (Fig 3). Statistically, Eggers's test result also depicted the absence of statistically significant publication bias (small studies effect) ($p = 0.484$).

**Sensitivity analysis.** The impact of individual studies on the pooled estimate of WMSD was evaluated by performing a sensitivity analysis. The finding revealed that none of the included studies affected the pooled estimate (Fig 4).

**Subgroup analysis.** Based on region category, the highest pooled prevalence of WMSD was recorded among studies conducted in other regions (67%, 95% CI: 55–79%) as compared to studies conducted in the Amhara region (48%, 95% CI: 18–77%) (Fig 5). Regarding the articles' publication year category, the highest pooled WMSD was observed among

**Table 1. characteristics of the included studies to estimate the pooled prevalence of WMSD among bank workers in Ethiopia, 2024.**

| Authors | Year of Publication | Region | Method of data collection | Study Design | Sample Size | Preva-lence (%) | Quality Score (%) |
|---|---|---|---|---|---|---|---|
| Dagne et al [23] | 2020 | Addis Ababa | Self-administration | Cross-sectional | 755 | 77.6 | 100 |
| Temesgen et al [29] | 2023 | Amhara | Self-administration | Cross-sectional | 266 | 63.2 | 75 |
| Workneh and Mekonen [26] | 2022 | Amhara | Self-administration | Cross-sectional | 285 | 55.4 | 87.5 |
| Jonga et al [28] | 2023 | SNNPR | Self-administration | Cross-sectional | 607 | 51.9 | 87.5 |
| Etana et al [25] | 2021 | Oromia | Interview | Cross-sectional | 335 | 73.1 | 100 |
| Kibret et al [24] | 2020 | Tigray | Interview | Cross-sectional | 307 | 65.5 | 100 |
| Demissie et al [27] | 2022 | Amhara | Interview | Cross-sectional | 422 | 61.1 | 100 |
| Demissie et al [30] | 2023 | Amhara | Interview | Cross-sectional | 422 | 11.7 | 87.5 |

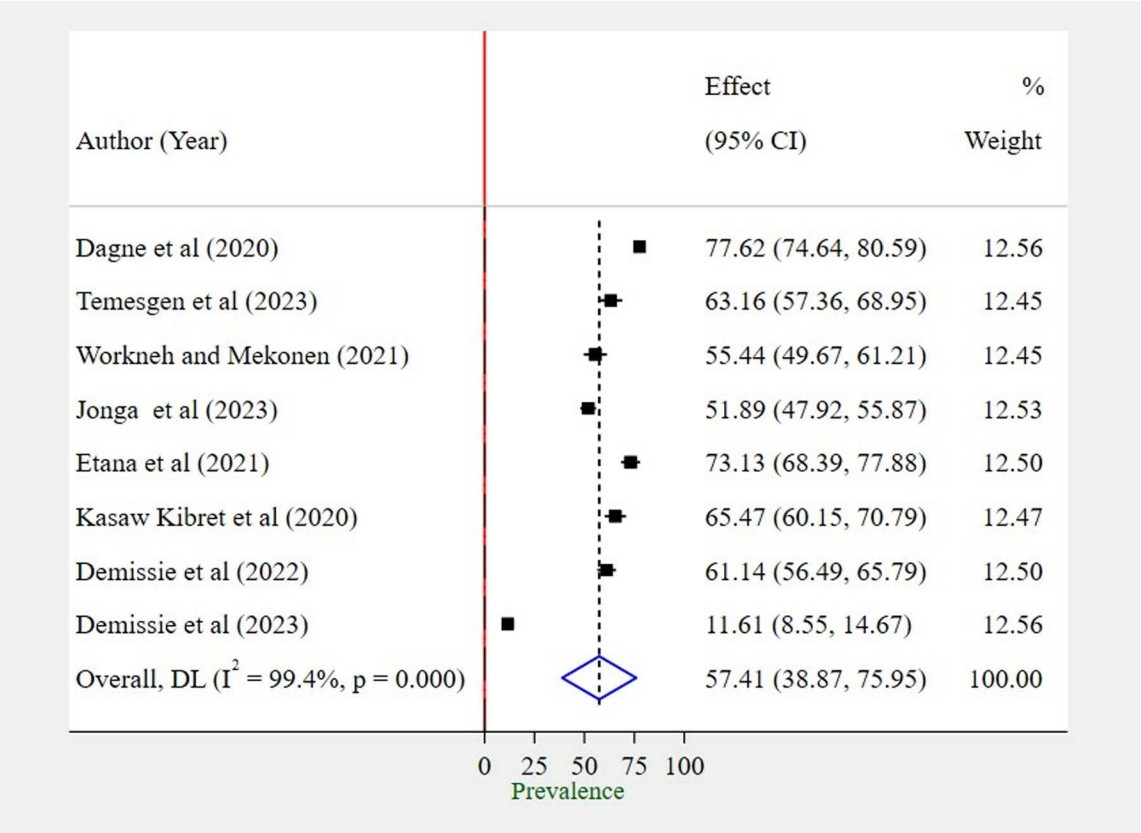

**Fig 2. Forest plot showing the pooled prevalence of WMSD among bank workers in Ethiopia, 2024.**

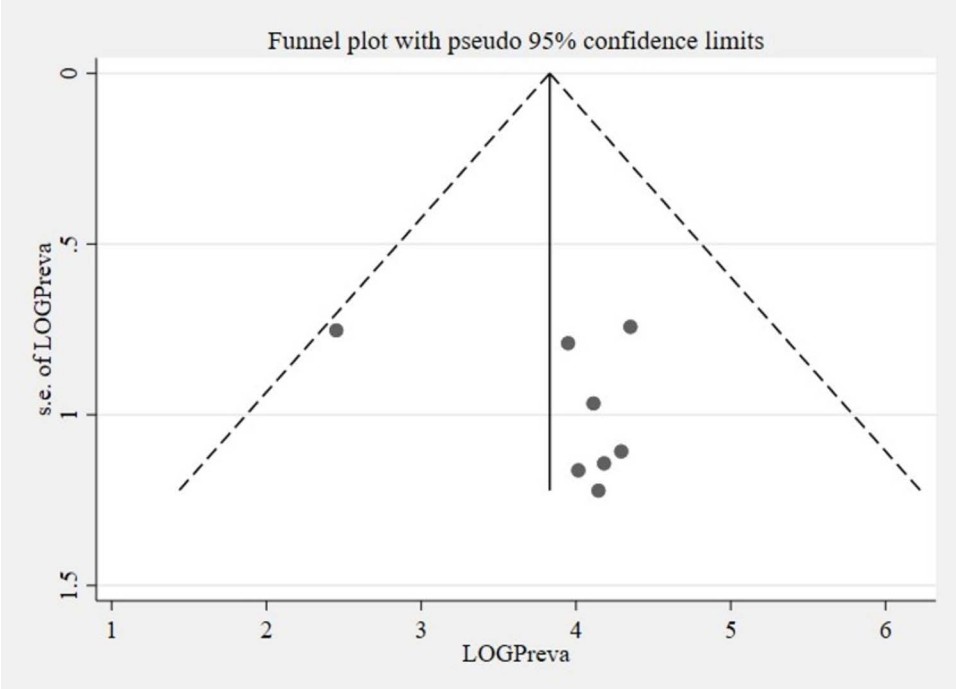

**Fig 3. A funnel plot to test the publication bias of the included studies of the meta-analysis.**

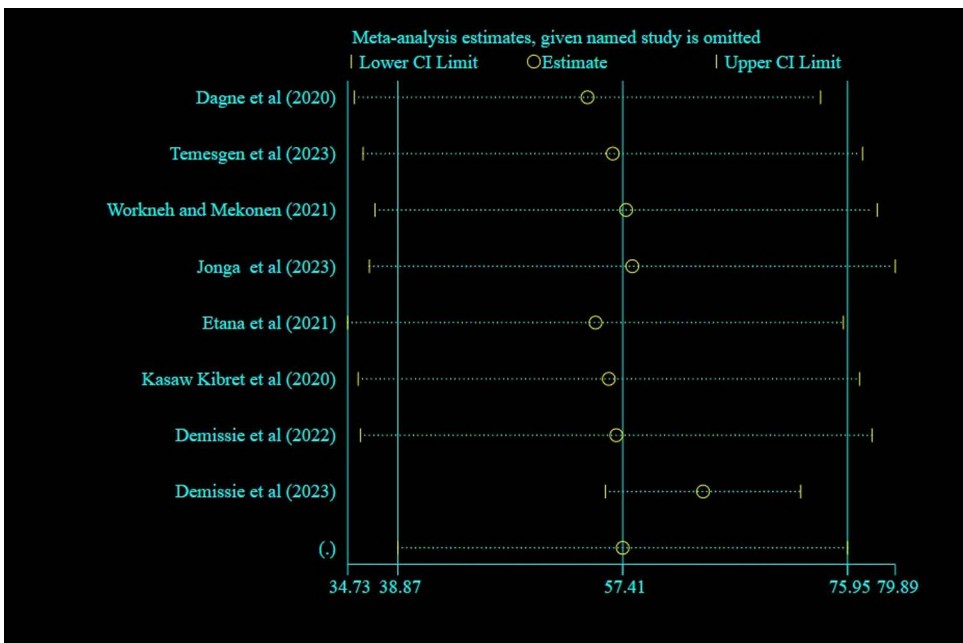

**Fig 4. A sensitivity analysis result of the included studies for WMSD among bank workers in Ethiopia, 2024.**

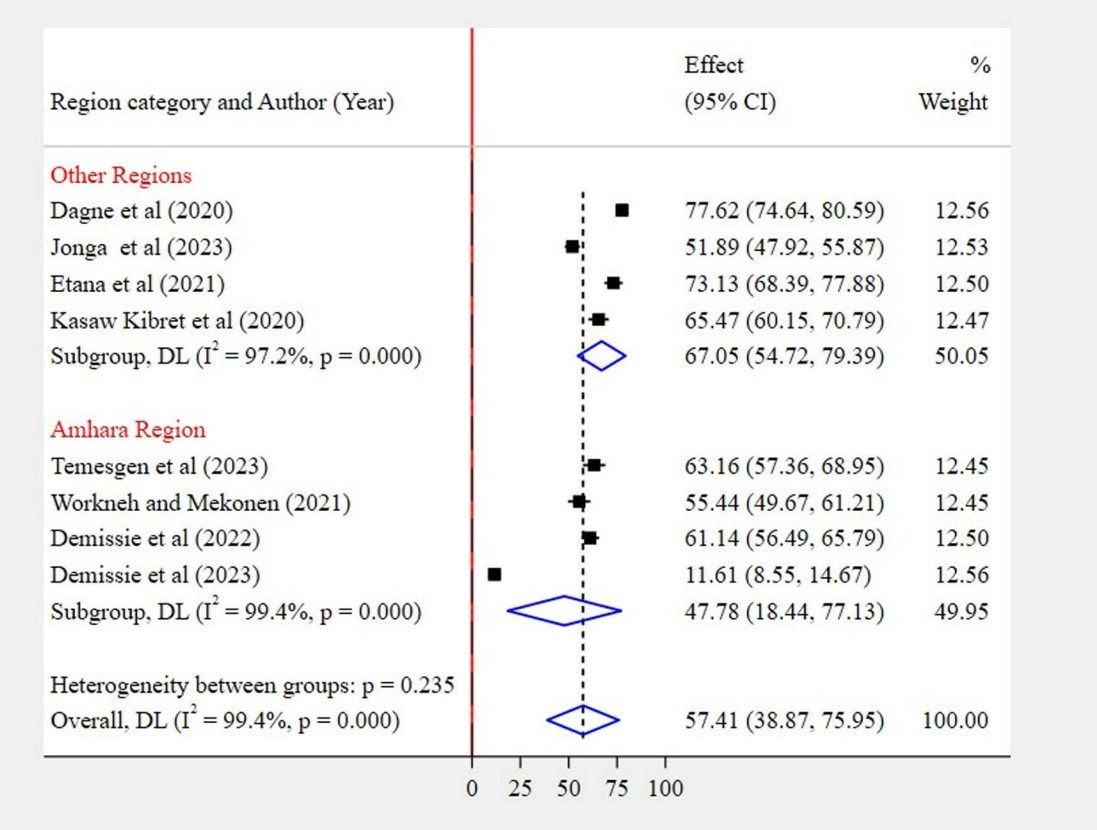

**Fig 5. Subgroup analysis by region category.**

studies published before 2022 (72%, 95% CI: 65–79%) as compared to those studies published in 2022 and after (49%, 95% CI: 26–71%) (Fig 6). The pooled prevalence of WMSD was higher among the studies having a sample size of 425 and above (68%, 95% CI: 59–77%) as compared to those with lower than 425 (47%, 95% CI: 20–74%) (Fig 7). The highest pooled prevalence was recorded for those studies that used self-administration as a data collection method (62%, 95% CI: 48–76%) (Fig 8).

**Factors associated with work-related musculoskeletal disorder.** There were seventeen factors repeatedly presented in the included articles of this meta-analysis. The factors were gender, age (30–39 years, ≥40 years), BMI (underweight, overweight), type of sitting position (back-twisted, back-bent), worktime break, repetitive motion, type of chair, job stress, awkward/sideway reaching, physical activity, ergonomic training, work experience and duration of computer use per day (Table 2).

The association between gender and WMSD among the included five studies [23,26–28,30] has been assessed. The result showed that there was a significant association in three of the included studies. According to this meta-analysis, the odds of WMSD were two times higher among female bank workers as compared to their counterparts (POR = 2.03; 95% CI: 1.13–2.92) (Fig 9). Based on the findings of four studies, the association between WMSD and job stress was assessed [23–26]. In all of these studies, a positive association was found. According to the results of this meta-analysis, the odds of WMSD were 3 times higher among bank workers who had job stress as compared to those who had no job stress (POR = 3.09; 95% CI: 1.89–4.28) (Fig 10). Six articles [24–28,30] were included to identify the association between physical activity and WMSD. Four of the included studies had a significant association. Based on the results of the

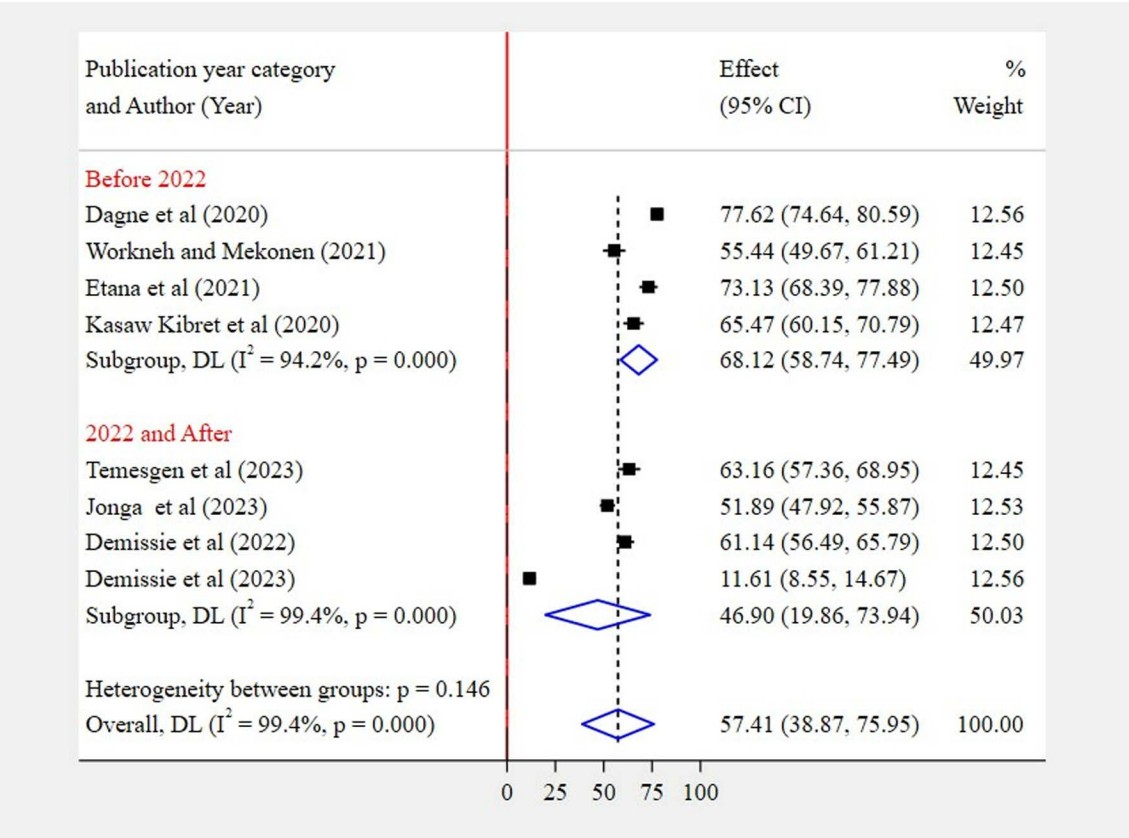

**Fig 6. Subgroup analysis by publication year category.**

meta-analysis, it was revealed that the odds for the occurrence of WMSD among bank workers who did not experience physical activity were 3 times higher when compared to those bank workers who did (POR = 3.13; 95% CI: 1.62–4.65) (Fig 11). Similarly, two articles [24,30] were included to determine the link between work experience and WMSD. In this case, both articles had a positive and significant association with WMSD. The results of this meta-analysis revealed that the odds for the occurrence of WMSD among bank workers who had five years and above work experience were 5 times higher as compared to those who had less than five years of work experience (POR = 4.78; 95% CI: 1.48–8.09) (Fig 12).

## Discussion

Work-related musculoskeletal disorders have continued to be a global burden to have a substantial impact on the health and productivity of workers in occupational settings. So far, many studies have been conducted to identify the possible risk factors associated with WMSD. However, these findings were not found to be consistent and conclusive enough. This in turn hinders the efforts of effective and timely intervention activities. Therefore, the current systematic review and meta-analysis aimed to determine the pooled prevalence of WMSD and its associated factors among bank workers in Ethiopia was conducted.

The pooled prevalence of WMSD among bank workers in Ethiopia was found to be 57.41% (95%; CI: 38.87%, 75.95%). This figure is in line with a global systematic review and meta-analysis of pooled prevalence reports on musculoskeletal disorders at the lower back among operating room personnel (61.48%) [39] and the prevalence of musculoskeletal disorder among healthcare professionals in Africa [40]. Additionally, the pooled prevalence is congruent with a

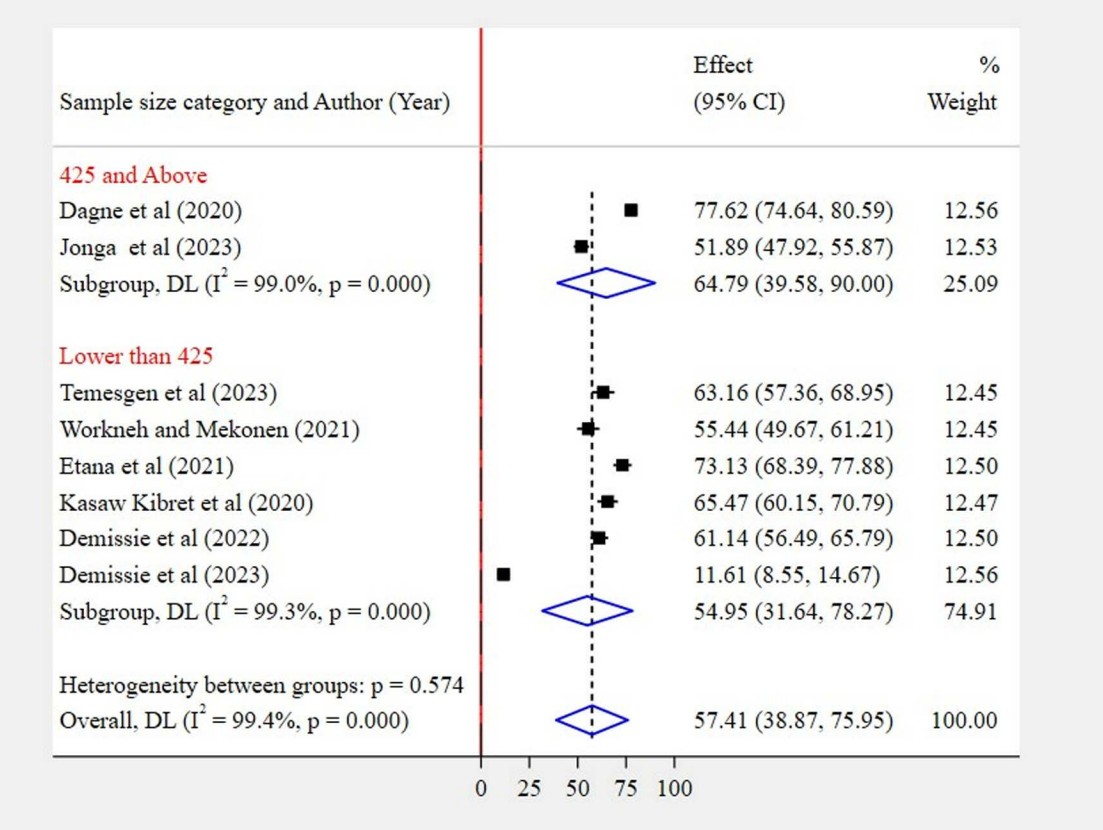

**Fig 7. Subgroup analysis by sample size category.**

systematic review and meta-analysis done among workers in the automobile manufacturing industry in China (53.1%) [41] and other studies conducted among electronic manufacturing workers in China (40.6%) [42], office workers in Lebanon (45.2%) [43], bank staff in Kigali, Rwanda (45.8%) [44], healthcare providers working in the operation room in Ethiopia (64.2%) [45] and higher education institutions' office workers in Ethiopia (71.9%) [46].

However, the figure is higher than a systematic review and meta-analysis report on the occupational- related pooled prevalence of upper extremity musculoskeletal pain at the elbow part of the body among the working population of Ethiopia (33.7%) [47]. Moreover, the report is higher than the prevalence of a study conducted among workers in Taiwan (37%) [48], dentists in India (34.5%) [49], and automobile manufacturing production workers in Korea (27.4%) [50]. The availability and quality of centers for physical exercise, job rotation, provision of psychosocial training, and workload reduction might explain the discrepancy.

The odds of WMSD were two times higher among females as compared to their counterparts. This result is in line with the findings of the studies conducted in India [51], China [41], and Kuwait [52]. The most likely explanation for this might be the hormonal, somatic, and psychological aspects following the difference in gender as evidenced by Gagnon et al [53], in which the structure of muscle and ligaments soft tissue at the lower back or waist are weaker for females than that of males. The odds of WMSD among bank workers who had job stress were three times higher as compared to those who had no job stress. This result is congruent with the findings of the studies conducted in the Hunan province of China [54], Switzerland [55], and Nigeria [56]. The possible justification might be evidenced by the fact that work-stress-induced psychological burdens become heavier, which in turn intensifies muscle strain [57,58]. Implementing practical stress

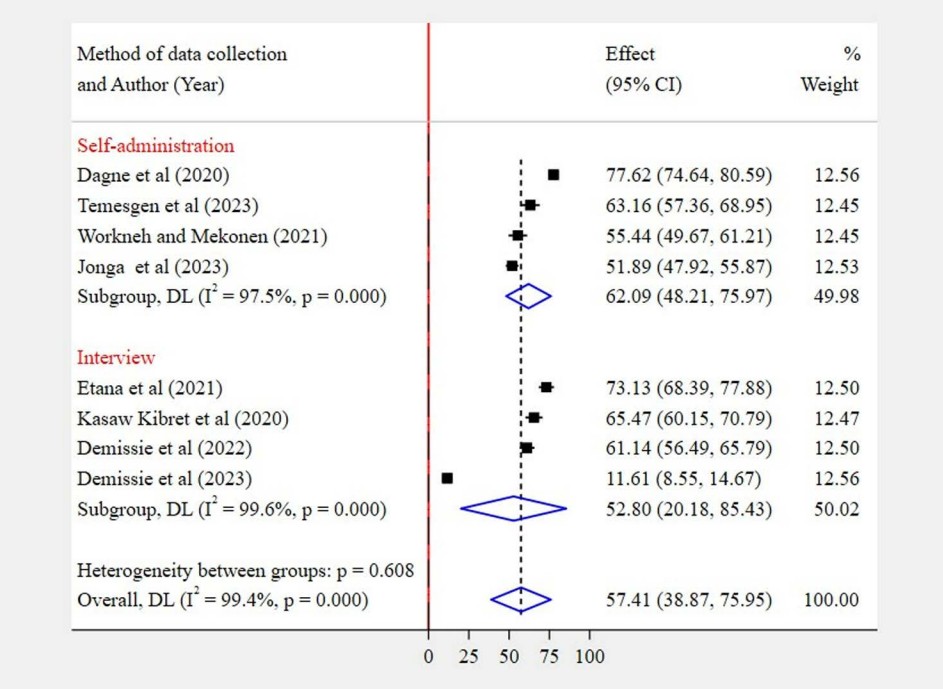

**Fig 8. Subgroup analysis by the method of data collection.**

management measures are pivotal and recommended as a study conducted in Ethiopia is in support of this, and high-lighted the importance of organizational stress management strategies such as eliminating hazards or minimizing employees' exposure to them, enhancing the organization's capacity to identify and address work-related issues, and providing support to help employees cope and recover [59].

The odds of WMSD among bank workers who did not experience physical activity were three times higher as compared to those who did. This is congruent with the evidence from China [60], which elucidates that workers who did not undertake physical exercise showed an elevated risk of WMSD. Moreover, it is evidenced that workplace physical activity programs have shown promise in promoting occupational health and reducing both stress and musculoskeletal pain among employees. By incorporating physical activity into the workday, these programs can improve overall health, decrease discomfort, and enhance the ability to perform job tasks [61,62]. Regular physical activity offers a protective effect against the development of musculoskeletal discomfort and injuries. Engaging in physical activity for twenty minutes three times a week can help alleviate discomfort in various areas, including the shoulders, neck, and lower back [63–65]. Participants who engaged in more frequent exercise sessions reported significant improvements in overall well-being. They were 74% more likely to experience psychophysiological well-being, 30% less likely to encounter difficulty in performing tasks, and a remarkable 87% more likely to perceive enhanced interpersonal relationships. These findings underscore the profound impact of physical activity on both physical and mental health, highlighting its potential to improve overall quality of life [61]. Additional evidence from a randomized controlled trial demonstrated the positive effects of workplace exercise on mood, performance, and overall well-being. Participants experienced improved concentration, enhanced problem-solving skills, a clearer mind, renewed energy, strengthened work relationships, and increased resilience to stress. Moreover, exercising provided an opportunity for interaction and connection with colleagues, fostering a more positive and collaborative work environment [66].

**Table 2. The pooled odds ratio for factors associated with WMSD among bank workers in Ethiopia, 2024.**

| Listed variables | | Number of study participants | Number of studies included | Pooled Odds Ratio (95% CI) | Heterogeneity | |
|---|---|---|---|---|---|---|
| | | | | | I² | p-value |
| Gender | Male | 1 | 1 | 1 | 1 | 1 |
| | Female | 2491 | 5 | 2.03(1.13-2.92) | 62.1% | 0.032 |
| Age | < 30 Years | 1 | 1 | 1 | 1 | 1 |
| | 30-39 Years | 2004 | 4 | 1.50(0.55_2.45) | 22.6% | 0.275 |
| | ≥ 40 Years | 2004 | 4 | 2.60(0.71-4.48) | 0.0% | 0.914 |
| Smoking behavior | Yes | 1177 | 2 | 3.78(0.17-7.38) | 0.0% | 0.682 |
| | No | 1 | 1 | 1 | 1 | 1 |
| BMI | Normal | 1 | 1 | 1 | 1 | 1 |
| | Underweight | 1090 | 2 | 0.90(0.37-1.44) | 0.0% | 0.607 |
| | Overweight | 1090 | 2 | 0.89(0.38-1.40) | 0.8% | 0.315 |
| Type of sitting position | Right position | 1 | 1 | 1 | 1 | 1 |
| | Back-twisted | 1040 | 2 | 2.22(0.73-5.17) | 68.6% | 0.074 |
| | Back-bent | 1040 | 2 | 2.25(0.96-5.46) | 88.8% | 0.003 |
| Worktime break | Yes | 1 | 1 | 1 | 1 | 1 |
| | No | 1484 | 3 | 2.21(0.03-4.46) | 53.6% | 0.116 |
| Repetitive motion | Yes | 1512 | 3 | 1.57(0.51-2.62) | 58.6% | 0.089 |
| | No | 1 | 1 | 1 | 1 | 1 |
| Type of chair | Fixed | 1728 | 4 | 1.73(0.31-3.77) | 78.8% | 0.003 |
| | Not fixed | 1 | 1 | 1 | 1 | 1 |
| Job stress | Yes | 1682 | 4 | 3.09(1.89-4.28) | 0.0% | 0.487 |
| | No | 1 | 1 | 1 | 1 | 1 |
| Awkward/sideway reaching | Yes | 601 | 2 | 2.53(0.23-4.83) | 63.7% | 0.097 |
| | No | 1 | 1 | 1 | 1 | 1 |
| Physical activity | Yes | 1 | 1 | 1 | 1 | 1 |
| | No | 2378 | 6 | 3.13(1.62-4.65) | 83.9% | <0.001 |
| Ergonomic training | Yes | 1 | 1 | 1 | 1 | 1 |
| | No | 1758 | 4 | 3.65(0.18-7.12) | 94.6% | <0.001 |
| Work experience | < 5 years | 1 | 1 | 1 | 1 | 1 |
| | ≥ 5 years | 729 | 2 | 4.78(1.48-8.09) | 11.8% | 0.287 |
| Duration of computer use per day | < 8 hours | 1 | 1 | 1 | 1 | 1 |
| | ≥ 8 hours | 844 | 2 | 2.57(0.25-7.39) | 93.1% | <0.001 |

1=Reference category.

The odds of WMSD among bank workers who had five years and above work experience were five times higher as compared to those who had less than five years of work experience. This result is in line with the findings of a study conducted in China [42], which reported that a higher risk of developing WMSD was observed among workers who had work experience of five years and above as compared to those who had less work experience. The possible reason for this might be the development of cumulative trauma or repetitive strains due to the long time which in turn attributed to the development of WMSD. Additionally, the result of this study is in line with the finding of another study conducted in Iran which indicated longer work experience increases the odds of WMSD [67]. However, this result is contrary to the findings of a study

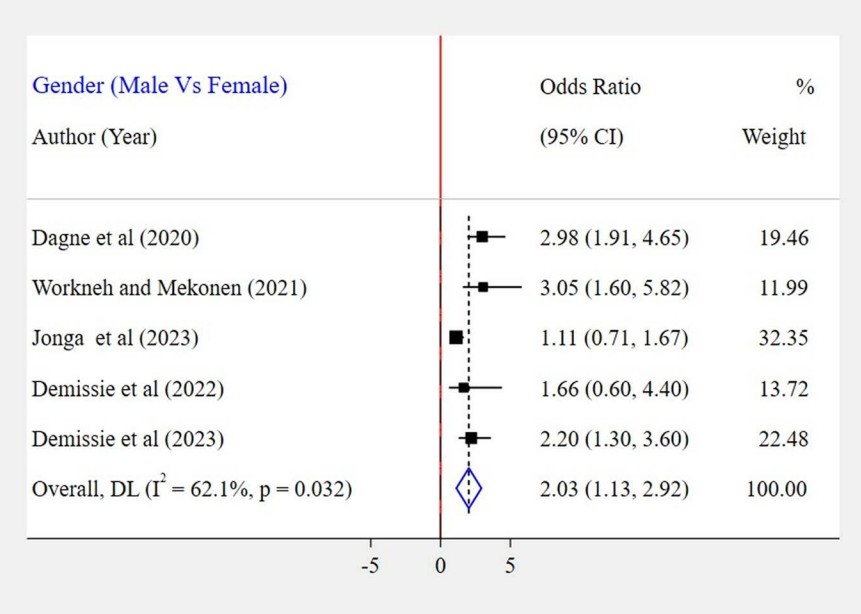

**Fig 9. A forest plot of odds ratio showing the association between gender and WMSD among bank workers in Ethiopia, 2024.**

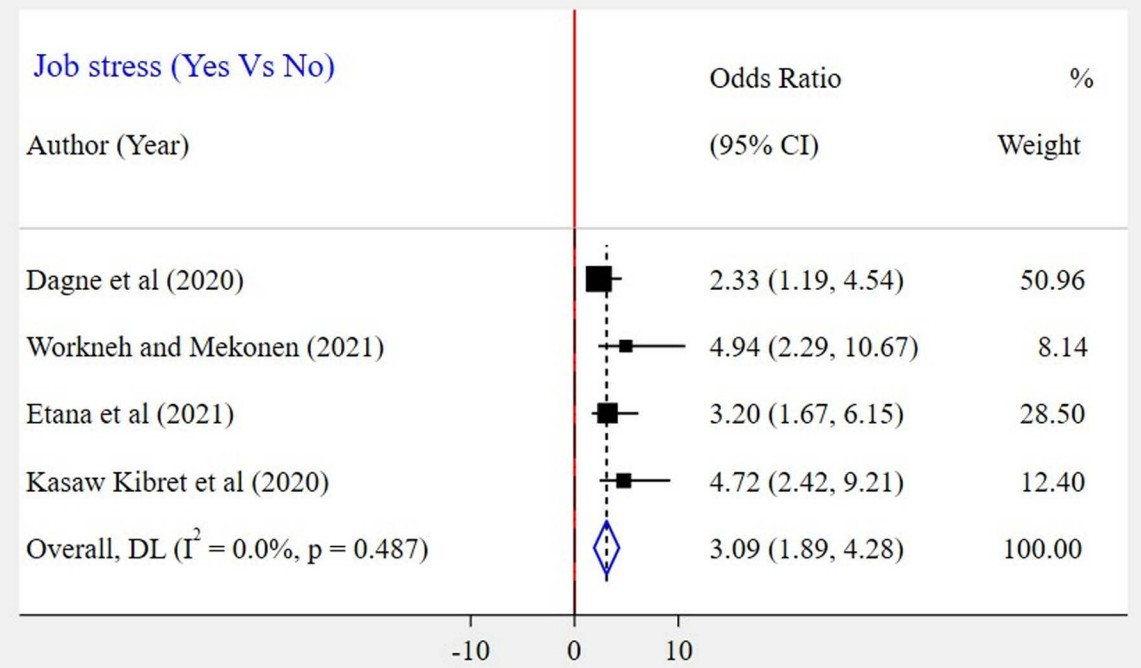

**Fig 10. A forest plot of odds ratio showing the association between job stress and WMSD among bank workers in Ethiopia, 2024.**

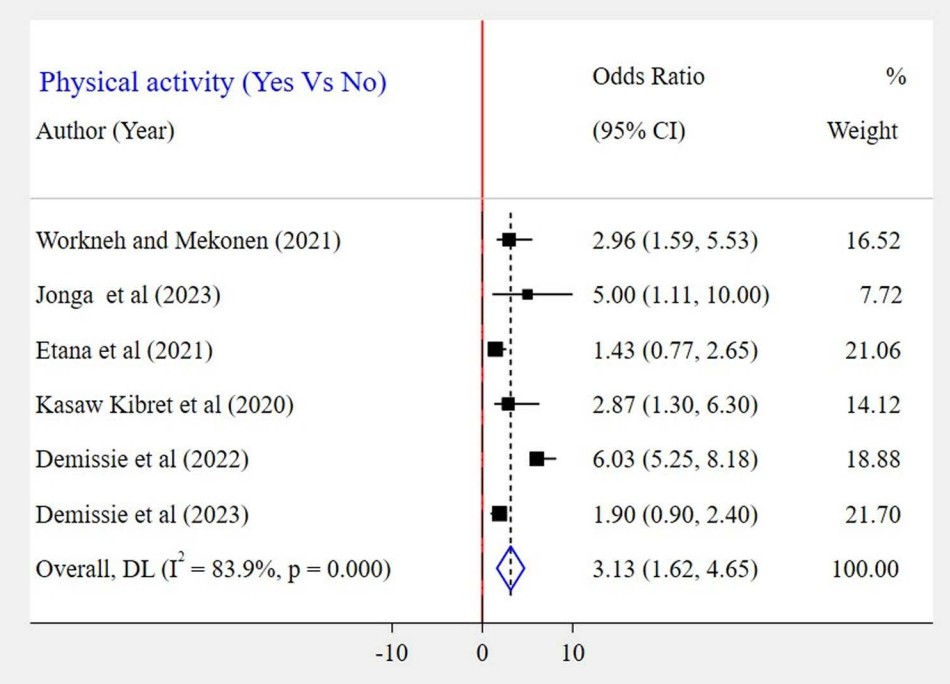

**Fig 11.  A forest plot of odds ratio showing the association of physical activity and WMSD among bank workers in Ethiopia, 2024.**

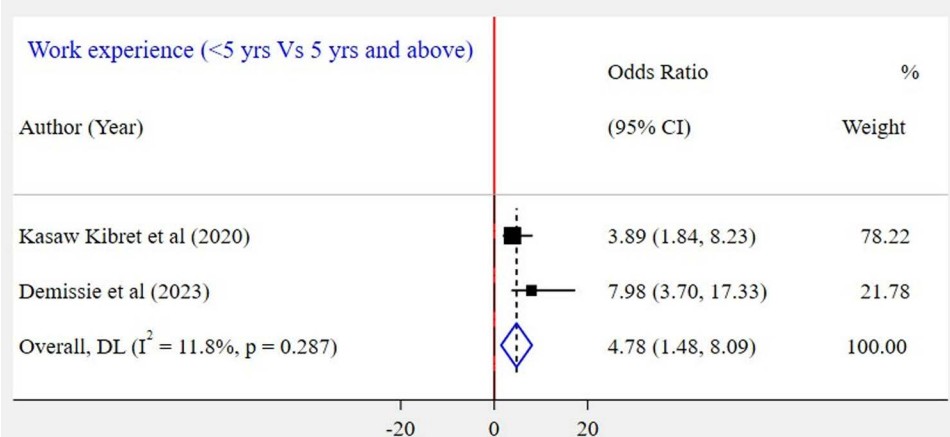

**Fig 12.  A Forest plot of odds ratio showing the association between work experience and WMSD among bank workers in Ethiopia, 2024.**

conducted in Nigeria [56], which revealed that low work experience is one of the workplace factors that increase the odds of WMSDs. This might be due to the growing awareness to prevent and control risk factors as workers get more experience.

## Limitations of the study

Even though the systematic review and meta-analysis was conducted based on the latest PRISMA guideline, it did not consider all the regions in Ethiopia due to a lack of availability of articles. Moreover, due to the limited number of studies with the topic of interest in Ethiopia, the pooled prevalence was estimated with only eight articles.

## Conclusion

In this study, the recorded pooled prevalence of work-related musculoskeletal disorder among bank workers in Ethiopia was high. Female gender, absence of physical activity, presence of job stress, and having work experience of five years and above were the factors significantly associated with work-related musculoskeletal disorder. Even though the effectiveness and timeliness of different intervention methods for work-related musculoskeletal disorders need further research, building centers for physical exercise, provision of psychosocial training, reduction of job-related stresses, and reduction of workload by the government, bank institutions, federal ministry of health (FMOH), and other partners and stakeholders are pivotal in preventing and reducing work-related musculoskeletal disorder in occupational areas such as banks. Furthermore, implementing fundamental occupational health and safety measures in office settings is strongly recommended. These include raising awareness, providing adequate facilities, ensuring supportive management, minimizing repetitive tasks, optimizing workspace layout, and conducting regular inspections.

## Supporting information

**S1 Table. PRISMA 2020 checklist.**
(DOCX)

**S2 Table. Results of JBI quality assessment.**
(DOCX)

**S3 Table. All articles excluded and included in the study.**
(XLSX)

**S4 Table. Data extraction sheet.**
(XLSX)

## Author contributions

**Conceptualization:** Abebe Kassa Geto, Endalew Minwuye Andargie, Birhanu Sewunet.

**Data curation:** Abebe Kassa Geto, Birhanu Sewunet, Tarikuwa Natnael, Chala Daba.

**Formal analysis:** Abebe Kassa Geto, Chala Daba.

**Funding acquisition:** Abebe Kassa Geto, Hussien Mekonnen, Tesfalem Tilahun Yemane, Endalew Minwuye Andargie.

**Investigation:** Abebe Kassa Geto, Hussien Mekonnen, Birhanu Sewunet.

**Methodology:** Abebe Kassa Geto, Tesfalem Tilahun Yemane.

**Project administration:** Abebe Kassa Geto, Tesfalem Tilahun Yemane, Birhanu Sewunet, Chala Daba.

**Resources:** Abebe Kassa Geto, Hussien Mekonnen, Endalew Minwuye Andargie, Tarikuwa Natnael.

**Software:** Abebe Kassa Geto, Chala Daba.

**Supervision:** Abebe Kassa Geto, Tesfalem Tilahun Yemane, Endalew Minwuye Andargie, Tarikuwa Natnael.

**Validation:** Abebe Kassa Geto, Endalew Minwuye Andargie, Birhanu Sewunet, Chala Daba.

**Visualization:** Abebe Kassa Geto, Hussien Mekonnen, Birhanu Sewunet.

**Writing – original draft:** Abebe Kassa Geto, Birhanu Sewunet.

**Writing – review & editing:** Abebe Kassa Geto, Hussien Mekonnen, Tesfalem Tilahun Yemane, Endalew Minwuye Andargie, Tarikuwa Natnael, Chala Daba.

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
