## [Decision Letter · Decision Letter 0]

26 Dec 2024

PONE-D-24-47282Work-related Musculoskeletal disorder and its associated factors among bank workers in Ethiopia: A systematic review and meta-analysis.PLOS ONE

Dear Dr. Geto,

Thank you for submitting your manuscript to PLOS ONE. After careful consideration, we feel that it has merit but does not fully meet PLOS ONE’s publication criteria as it currently stands. Therefore, we invite you to submit a revised version of the manuscript that addresses the points raised during the review process.

**ACADEMIC EDITOR: **Dear Authors,Your manuscript was reviewed by me and two expert reviewers. While both reviewers find merit in your manuscript, they have raised several important feedback and comments. I also believe that those comments are very useful to improve the quality of the manuscript. You should revise your manuscript based on reviewer and editor comments.Comments:1. There are many grammatical errors throughout the manuscript. I would suggest for a language editing by a native speaker.2. Authors should add statistics to support their claim in the abstract.3. Please confirm if risk of bias was assessed? how?4. Authors stated that the article was searched by 4 reviewers, screened by 3 reviewers, and the quality was assessed by 2 reviewers. Why were all reviewers not utilized in all the stages? Whether inter-reviewer reliability was tested in any stages? what was the results?5. in the result section, authors should add comparators for all listed variables in Table 2. 

We look forward to receiving your revised manuscript.

Kind regards,

Shahnawaz Anwer, PhD

Academic Editor

PLOS ONE

Journal Requirements:

2. As required by our policy on Data Availability, please ensure your manuscript or supplementary information includes the following: 

Reviewers' comments:

Reviewer's Responses to Questions

**Comments to the Author**

1. Is the manuscript technically sound, and do the data support the conclusions?

Reviewer #1: Yes

Reviewer #2: Yes

2. Has the statistical analysis been performed appropriately and rigorously? 

Reviewer #1: Yes

Reviewer #2: Yes

3. Have the authors made all data underlying the findings in their manuscript fully available?

Reviewer #1: Yes

Reviewer #2: Yes

4. Is the manuscript presented in an intelligible fashion and written in standard English?

Reviewer #1: Yes

Reviewer #2: Yes

5. Review Comments to the Author

Reviewer #1: MANUSCRIPT ID: PONE-D-24-47282

MANUSCRIPT TITLE: Work-related Musculoskeletal disorder and its associated factors among bank workers in Ethiopia: A systematic review and meta-analysis

REVIEWER COMMENT

The manuscript entitled “Work-related Musculoskeletal disorder and its associated factors among bank workers in Ethiopia: A systematic review and meta-analysis”. The information provided in this manuscript is beneficial for researchers and academia, but this study has some limitation as the provided information is only for the pooled prevalence of and factors associated with work-related musculoskeletal disorder among bank workers in Ethiopia; and I have suggested some revisions.

Abstract: results- The study reveals in terms of the gender, job stress, physical activity, and work experience were found to be factors significantly associated with work-related musculoskeletal disorder why and what measure the author taken to end up such conclusion should be stated more in the abstract. Research on other variables is limited- please elaborate other variables related to musculoskeletal disorder.

Systematic review and meta-analysis - The author mentioned in statistical analysis table-1, all the study are cross sectional, do the study find any longitudinal study based on Ethiopia and or greater Africa if not then stated in the methods. the result table-2 BMI classification only two class overweight and underweight what about normal BMI range data- Do the normal BMI has any relation with work-related musculoskeletal disorder.

Results: Could the author explain more how they categorized low risk bias, unclear risk of bias.

• Please avoid repetition-

• Please check reference style throughout MS

• Recheck Legends description is as per figure number and discussion.

Reviewer #2: This manuscript provides a thorough examination of the pooled prevalence and associated factors of work-related musculoskeletal disorders (WMSDs) among bank workers in Ethiopia, addressing a critical occupational health issue. However, several areas require refinement to enhance the clarity, rigor, and impact of the study. The keywords should be formatted according to MeSH (Medical Subject Headings) terms for improved indexing.

The methodology adheres to the PRISMA guidelines but would benefit from a more detailed explanation of how heterogeneity across studies was addressed. While statistical tools, such as I², were used, discussing additional steps taken to minimize variability, such as subgroup analyses or sensitivity tests, would strengthen the robustness of the study. A clearer justification for the use of the random-effects model and a more detailed description of the criteria for including and excluding studies, particularly gray literature, are also recommended.

The discussion contextualizes the findings well within global and regional studies but could better highlight how these findings can inform occupational health interventions and policies in Ethiopia. Discussing practical recommendations, such as workplace ergonomic improvements or stress management programs, will enhance the applicability of the manuscript.

Additionally, the conclusion should focus more on the implications of the findings for occupational health policies and the need for further research to include diverse regions and adopt a longitudinal design.

Figures and tables are informative but lack clear legends and captions, making them less accessible to the readers. Ensure that all visual elements are self-contained with sufficient explanatory details. Finally, a proofreading pass is required to address minor grammatical inconsistencies and ensure clarity throughout the text.

6. PLOS authors have the option to publish the peer review history of their article (what does this mean? ). If published, this will include your full peer review and any attached files.

**Do you want your identity to be public for this peer review?** For information about this choice, including consent withdrawal, please see our Privacy Policy .

Reviewer #1: **Yes: ** RUHINA BINAT GHANI

Reviewer #2: No

---

## [Author Response · Author response to Decision Letter 1]

8 Mar 2025

Date: 4 March 2025

To Esteemed Editor/s and Reviewers

Subject: Response to Editors’ and Reviewers’ Comments and Suggestions

It is our great pleasure to thank the editors and reviewers of our manuscript entitled “Work-related Musculoskeletal disorder and its associated factors among bank workers in Ethiopia: A systematic review and meta-analysis” for the valuable comments and/or suggestions. Upon conducting a comprehensive review of the comments, suggestions and questions provided and asked, we have meticulously addressed each one. We are confident that the revisions we’ve made meet your expectations and significantly improve the quality of our manuscript. To aid in the review process, we have prepared two versions of the manuscript for your convenience: one with clear, easy to read copy and the other highlighted with track-changes showing the changes made from previously submitted version of the manuscript. These versions aim to facilitate understanding and transparency regarding the modifications made. We are pleased to present the revised manuscript along with our responses to the comments. We eagerly await your favorable consideration for publication.

We upload a point by point response word file to the given comments.

With Regards!

Authors

---

## [Editor Report · Decision Letter 1]

17 Apr 2025

Work-related Musculoskeletal disorder and its associated factors among bank workers in Ethiopia: A systematic review and meta-analysis.

PONE-D-24-47282R1

Dear Dr. Geto,

We’re pleased to inform you that your manuscript has been judged scientifically suitable for publication and will be formally accepted for publication once it meets all outstanding technical requirements.

Kind regards,

Shahnawaz Anwer, PhD

Academic Editor

PLOS ONE
---

## [Editor Report · Acceptance letter]

PONE-D-24-47282R1

PLOS ONE

Dear Dr. Geto,

I'm pleased to inform you that your manuscript has been deemed suitable for publication in PLOS ONE. Congratulations! Your manuscript is now being handed over to our production team.

Kind regards,

on behalf of

Dr. Shahnawaz Anwer

Academic Editor

PLOS ONE